# Focus, Distinguish, and Prompt: Unleashing CLIP for Efficient and Flexible Scene Text Retrieval

## ABSTRACT

Scene text retrieval aims to find all images containing the query text from an image gallery. Current efforts tend to adopt an Optical Character Recognition (OCR) pipeline, which requires complicated text detection and/or recognition processes, resulting in inefficient and inflexible retrieval. Different from them, in this work we propose to explore the intrinsic potential of Contrastive Language-Image Pre-training (CLIP) for OCR-free scene text retrieval. Through empirical analysis, we observe that the main challenges of CLIP as a text retriever are: 1) limited text perceptual scale, and 2) entangled visual-semantic concepts. To this end, a novel model termed FDP (Focus, Distinguish, and Prompt) is developed. FDP first focuses on scene text via shifting the attention to text area and probing the hidden text knowledge, and then divides the query text into content word and function word for processing, in which a semantic-aware prompting scheme and a distracted queries assistance module are utilized. Extensive experiments show that FDP significantly enhances the inference speed while achieving better or competitive retrieval accuracy. Notably, on the IIIT-STR benchmark, FDP surpasses the state-of-the-art method by 4.37% with a 4 times faster speed. Furthermore, additional experiments under phrase-level and attribute-aware scene text retrieval settings validate FDP's particular advantages in handling diverse forms of query text.

## CCS CONCEPTS

• **Information systems → Multimedia and multimodal retrieval**.

## KEYWORDS

Scene Text Retrieval, CLIP, Visual-Semantic Entanglement, Prompt Tuning

**ACM Reference Format:**
Anonymous Authors. 2024. Focus, Distinguish, and Prompt: Unleashing CLIP for Efficient and Flexible Scene Text Retrieval. In *Proceedings of the 32nd ACM International Conference on Multimedia (MM'24), October 28-November 1, 2024, Melbourne, Australia.* ACM, New York, NY, USA, 9 pages. https://doi.org/XXXXXXX.XXXXXXX

## 1 INTRODUCTION

Since text is ubiquitous in natural scenes and conveys rich semantic information, scene text understanding has received a lot of attention

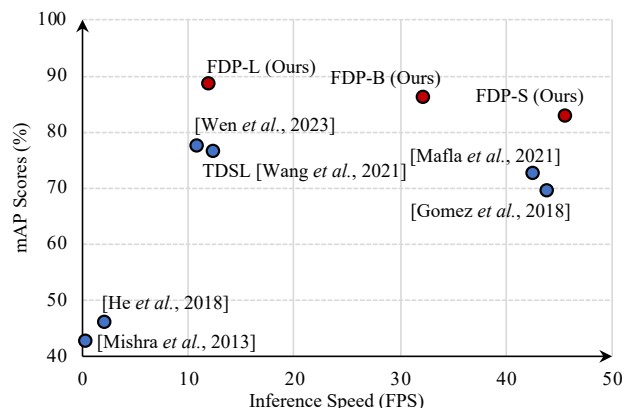

**Figure 1: Illustration of the trade-off between retrieval accuracy (mAP scores) and inference speed (FPS) on the IIIT-STR benchmark. Our proposed FDP method achieves better balance than previous methods.**

for decades. Different from common scene text understanding tasks such as text detection, text recognition, and end-to-end text spotting, Scene Text Retrieval (STR) is an emerging topic that only focuses on text of interest, *i.e.*, searching images containing a given query text from an image gallery. As such, STR is beneficial for many applications like product image search, program recommendation, and electronic book archives management [6, 7, 31].

With the aid of Optical Character Recognition (OCR) techniques, STR has made remarkable progress in recent years [9, 13, 16]. Nevertheless, existing methods still suffer from two critical limitations. First, as illustrated in Fig.1, there is a dilemma of how to balance retrieval accuracy (mAP scores) and inference speed (FPS). Specifically, most STR models follow the two-stage pipeline that first detects text regions and then compares these regions with the query text for retrieval. In this pipeline, either an exact text detection or recognition process is required, which significantly slows down the inference speed. Comparatively, Gomez *et al.* [8] achieve fast text retrieval using a single-shot CNN architecture, but it is limited by relatively low retrieval accuracy. Second, in real life, the query text that people expect to retrieve is often in various forms. However, current efforts rely on the local retrieval mechanism that treats word instances as query units, leading to inherent inflexibility in phrase-level or attribute-aware scene text retrieval (see Fig.2).

Recently, Contrastive Language-Image Pre-training (CLIP) [23] has become a powerful foundation model for learning cross-modal representations and enabling zero-shot transfer to downstream tasks. More remarkably, several works [14, 24] have demonstrated CLIP also implies OCR capabilities via pre-training on massive image-text pairs. It gives us a new insight: can we explore the intrinsic potential of CLIP for efficient and flexible STR? To this

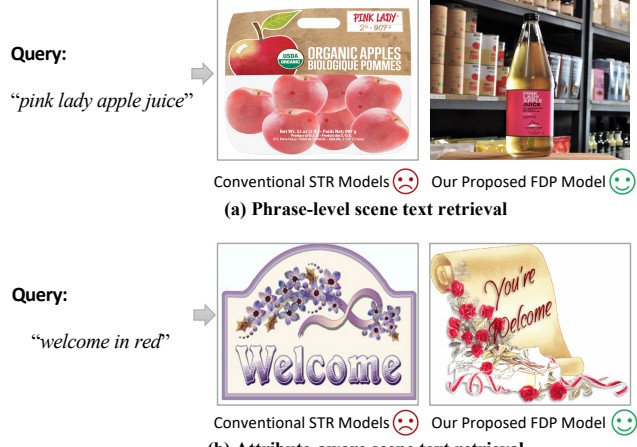

**(a) Phrase-level scene text retrieval**

**(b) Attribute-aware scene text retrieval**

**Figure 2: Illustration of the scene text retrieval in (a) phrase-level and (b) attribute-aware settings. Unlike conventional STR models that rely on the local retrieval mechanism, FDP is more flexible in handling diverse forms of query text.**

end, we investigate the advantages and deficiencies of CLIP in the STR task through an empirical study. A surprising finding is that simply applying the frozen CLIP can even achieve better accuracy than some dedicated STR models. Moreover, thanks to CLIP's simple network design, the retrieval speed is also superior. Despite these impressive results, there are still two challenges that hinder CLIP from being an ideal retrieval engine: 1) **Limited text perceptual scale.** As the image resolution input into CLIP is very limited (*e.g.*, 224×224), and scene text usually occupies only a small part of the scene image, a lot of text may be ignored or misrecognized by CLIP. 2) **Entangled visual-semantic concepts.** Due to the prevalence of text in natural images, there is confusion between visual text and semantic concepts in CLIP's cognition [20]. Its specific impact on STR is that the CLIP-based retrieval model performs much better on content words (*e.g.*, "*coffee*", "*hotel*") than on function words (*e.g.*, "*and*", "*with*") because only content words represent exact semantics. Besides, the model may have difficulty distinguishing similar words (*e.g.*, "*advice*" and "*advise*") because their semantics are close in the embedding space.

In this paper, we propose a model named FDP (Focus, Distinguish, and Prompt) to tackle the above challenges. Concretely, for each image in the gallery, we firstly force CLIP to **focus** on scene text by 1) applying the rough text localization results to refine the model attention on images, and 2) leveraging CLIP's well-aligned vision-language representations to prob text knowledge. Then, given a query text, we **distinguish** whether it is a content word or a function word via unsupervised clustering and determine the retrieval solution accordingly. Finally, a semantic-aware **prompt**ing scheme is developed, which converts the query text into a learnable prompt and ranks images by computing their similarity scores with each image. In addition, a distracted query assistance strategy is involved during training to resist the negative effects of similar words. Extensive experiments on three benchmarks show that FDP can achieve better or competitive accuracy compared to existing models with

a faster inference speed. To further evaluate the effectiveness of STR methods over arbitrary-length query text, we introduce a new benchmark of phrase-level scene text retrieval (PSTR). Meanwhile, qualitative experiments regarding attribute-aware scene text retrieval are conducted. These experimental results demonstrate the generalization and flexibility of FDP.

Overall, the main contributions of this work are three-fold:

1) To the best of our knowledge, it is the first work to directly extend CLIP for scene text retrieval. We summarize both the advantages and deficiencies of CLIP in dealing with the STR task and propose a novel FDP (Focus, Distinguish, and Prompt) method.

2) In contrast to previous works, FDP steers the prior knowledge from CLIP and eliminates the complicated text detection/recognition process, thus achieving a better trade-off between retrieval accuracy and inference speed. Notably, FDP outperforms the state-of-the-art method [30] by 4.37% mAP score with a 4 times faster speed on the IIIT-STR benchmark.

3) We evaluate existing STR methods in phrase-level and attribute-aware scene text retrieval settings, further verifying the superiority of FDP in handling diverse forms of query text.

## 2 RELATED WORK

### 2.1 Scene Text Retrieval

Most of the early STR approaches tend to follow the OCR pipeline [1, 26, 33]. They first take two separate steps of text detection and recognition to extract words in each image, and then match these words with the query word for retrieval. For instance, Mishra *et al.* [21] first investigate the STR task, proposing to rank all images based on the ordering and positioning of localized characters. Jaderberg *et al.* [11] perform text spotting with a CNN network and evaluate the occurrences of the query word within the spotted words. However, those straightforward attempts could not obtain satisfactory performance and are also not efficient. To solve this problem, Gomez *et al.* [8] leverage a compact representation named Pyramidal Histogram of Character (PHOC) [2] and propose a single-shot CNN architecture that simultaneously predicts text proposals and corresponding PHOCs. In this way, the STR task can be completed by a simple nearest neighbor search. Considering current handcraft representations (including PHOC) still cannot well reflect the distance between text and image modalities, recent methods are dedicated to mining better similarity measures. TDSL [27] establishes an end-to-end network that jointly optimizes text detection and cross-modal similarity learning. To mitigate the gap across different modalities, Wen *et al.* [30] propose to cast STR as an image-to-image matching problem. Although better retrieval accuracy is achieved, it comes at the cost of inference speed.

### 2.2 Exploring CLIP's OCR Capabilities

Vision-language models pre-trained on web-scale data have been demonstrated to exhibit certain OCR capabilities [10, 17, 18, 32, 35]. As reported in [23], the CLIP model shows favorable OCR performance in rendered text images. To further mine the underlying rationales, [14] thoroughly inspects different versions of CLIP. This work uncovers that CLIP suffers from severe text spotting bias because many captions in CLIP's training dataset tend to parrot the

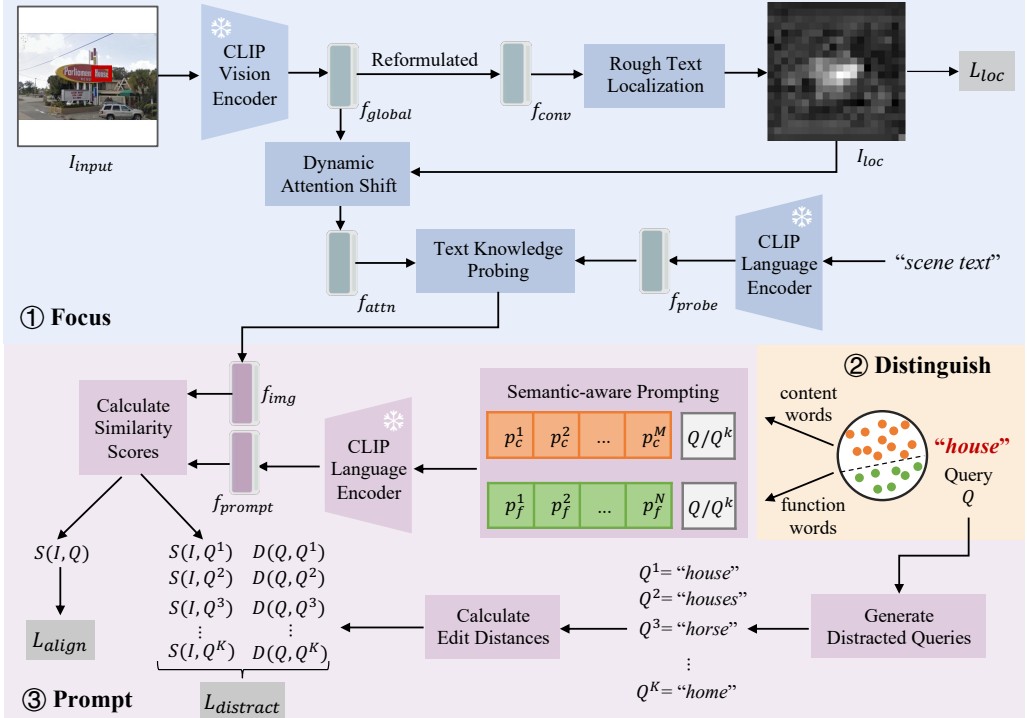

**Figure 3: The architecture of the proposed FDP model. It consists of three main parts: 1) Focus: Two main modules of dynamic attention shift and text knowledge probing are presented to highlight scene text information. 2) Distinguish: The query text is categorized into content words and function words via unsupervised clustering. 3) Prompt: The retrieval process is finally achieved by a semantic-aware prompting scheme, and meanwhile distracted queries are generated during training to assist in identifying similar words.**

visual text embedded within images. Through orthogonal projections of the learned representation space into "learn to spell" and "forget to spell" parts, [20] disentangles such bias to some extent. Besides, LoGoPrompt [24] finds that synthetic text images are good visual prompts for vision-language models, which can be used to improve image classification performance.

Inspired by these observations, several works aim to enhance OCR tasks by transferring the knowledge from CLIP. In the field of scene text recognition, CLIP4STR [36] designs a two-branch framework in which the recognition results are predicted by the visual branch and then refined by the cross-modal branch. CLIP-OCR [29] resorts to the knowledge distillation technique and guides the recognition with both visual and linguistic knowledge from CLIP. In the field of scene text detection, TCM [34] integrates CLIP with existing text detectors, leading to obvious performance improvements in domain adaptation and few-shot capabilities. However, CLIP merely acts as an auxiliary module in these works. Whether it is possible to turn CLIP directly into a scene text reader (spotter or retriever) remains an unexplored problem.

## 3 FDP METHOD

The overview of the proposed FDP framework is illustrated in Fig.3. Given a query text ($Q$ = "*house*"), FDP fulfills the STR task with a pipeline of "Focus, Distinguish, and Prompt".

### 3.1 Focus

Considering CLIP is pre-trained on conventional image-text pairs and thus lacks fine-grained awareness of visual text information, the first step of FDP is directing CLIP to focus on scene text. To be specific, for each image from the gallery, we first square it to obtain the input image $I_{input} \in \mathbb{R}^{L \times L}$ ($L$ is the image length), *i.e.*, perform zero-padding to make the shorter side match the longer side. The goal is to avoid the loss of image content caused by the center cropping operation during CLIP's preprocessing. Then, the frozen ResNet-based vision encoder of CLIP is employed to extract the global feature $f_{global} \in \mathbb{R}^{C \times H \times W}$ of $I_{input}$, where $C$, $H$ and $W$ stand for the channel, height and width dimensions respectively. Based on this global feature, two modules including dynamic attention shift and text knowledge probing are proposed to highlight scene text information and address the limited text perceptual scale problem.

**Dynamic Attention Shift.** The limitation of input resolution is an intractable problem encountered by pre-trained vision-language models. It greatly impairs scene text understanding performance because text often occupies a very small part of the image. Existing efforts resolve this problem by subdividing into image patches [12], retraining a vision encoder [3], or processing in the frequency domain [5], which are not efficient. Instead, in this work we find that it may be enough to give the model a glimpse of the rough area where text is clustered. To this end, we employ text detection

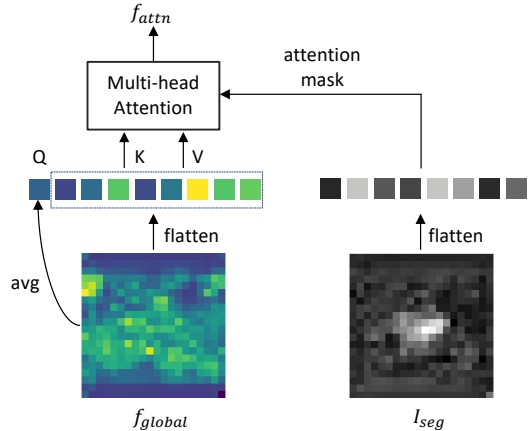

$f_{attn}$

attention mask

Multi-head Attention

Q  K  V

flatten  flatten

avg

$f_{global}$        $I_{seg}$

**Figure 4: Details of the dynamic attention shift module.**

supervision to train a lightweight text localization network, and then utilize the normalized probability map to reweigh the features in the average pooling layer. Specifically, as the multi-head attention layer in CLIP's vision encoder loses the 2D image information, we first introduce a reformulated head following [37] to recover the 2D convolutional image feature $f_{conv} \in \mathbb{R}^{E \times H \times W}$, where $E$ is the embedding dimension in CLIP. Then, the localization probability map $I_{loc} \in \mathbb{R}^{H \times W}$ is obtained via a learnable convolutional layer. We train the text localization network via a class-balanced cross-entropy loss, given by:

$$\mathcal{L}_{loc} = -\beta Y log(I_{loc}) - (1-\beta)(1-Y)log(1-I_{loc}) \quad (1)$$

where $Y$ is the ground-truth localization map generated by the text detection annotation, and $\beta$ is a balancing factor which is defined as:

$$\beta = 1 - \frac{\sum_{y \in Y} y}{|Y|} \quad (2)$$

After that, the predicted localization probability map is adopted as a new attention mask to dynamically refine the attention applied to the global feature. The details of the dynamic attention shift module are illustrated in Fig.4. CLIP uses Transformer-style multi-head attention to perform average pooling, where the 2D global feature is first flattened into a 1D sequence and then generates a key-value pair to interact with the globally average-pooled feature (query). Consequently, the localization probability map is also flattened into a 1D sequence and then weights the global feature at each spatial location. The derived attention feature $f_{attn} \in \mathbb{R}^E$ can shift the model attention to the scene text area.

**Text Knowledge Probing.** Empirically, we find that when we query CLIP with the word "*house*", it is possible to return the corresponding object (an image of a house) instead of the scene text (an image says "*house*"). This is because the neurons in CLIP's vision encoder tend to activate on the whole image rather than specific text information. Therefore, we consider whether we could design a simple strategy to probe the text-related knowledge hidden in CLIP. Drawing inspiration from previous work [23] that conducts zero-shot image classification using a predefined template "*a photo of [CLS]*", we propose to utilize the plain text "*scene text*" as a probe and obtain its language embedding as the probe feature $f_{probe} \in \mathbb{R}^E$,

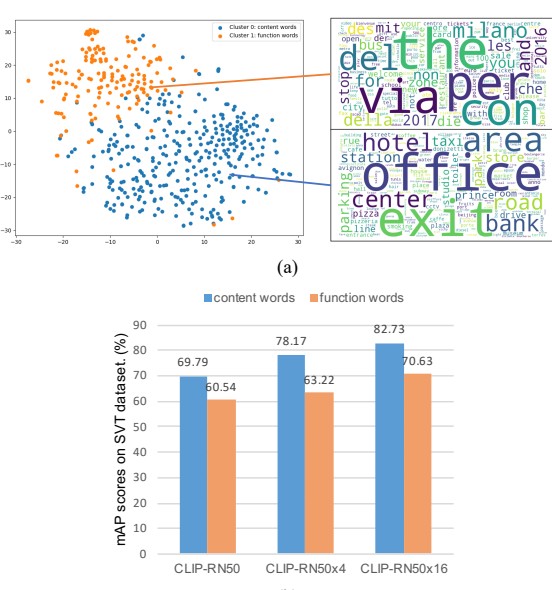

(a)

(b)

**Figure 5: Illustration of the effect caused by visual-semantic entanglement. (a) The t-SNE visualization of high-frequency scene text's CLIP language embeddings. (b) The comparison of the retrieval accuracy of three frozen CLIP models on content words and function words.**

which is then interacted with $f_{attn}$. Since the representations of vision and language are well-aligned in the embedding space of CLIP, this probe will naturally turn CLIP into a model that is more sensitive to scene text. Subsequently, the interacted feature is fused with the attention feature $f_{attn}$ as the image feature $f_{img}$ to comprehensively encode the image for retrieval. The text knowledge probing process is formulated as:

$$f_{img} = \text{MHCA}(Q = f_{attn}, K = f_{probe}, V = f_{probe}) + f_{attn} \quad (3)$$

where MHCA means the multi-head cross-attention mechanism.

### 3.2 Distinguish

Several works [14, 24] have revealed that the CLIP model exhibits inherent bias towards visual text, *e.g.*, an image of "dog" may be recognized as "cat" by placing the text that says "cat". The reason is that the captions CLIP pre-trained with are often simple repetitions of text embedded in images. In this work, we further observe that this bias is essentially the entanglement between visual and semantic concepts. To be specific, we select 500 words with the highest frequency from the MLT [22] training set, and then group their CLIP language embeddings into two clusters via K-Means. The t-SNE visualization result is depicted in Fig.5 (a). As can be seen, these words naturally fall into two clusters, namely, the content words and the function words. Between them, the content words have explicit semantics, usually appearing next to the thing they represent, so they exhibit strong visual-semantic entanglement. In contrast, the function words generally appear in the captions as conjunctions, which do not correspond to specific concepts. To

investigate this effect on STR, we evaluate the retrieval accuracy of content words and function words respectively by applying the frozen CLIP models on SVT [28] dataset, as shown in Fig.5 (b). It is obvious that for the three CLIP models with different capacities, the mAP scores on function words are significantly lower than those on content words. This inspires us that different retrieval solutions should be taken on these two clusters. Thus, before each retrieval process, we pre-distinguish whether the given query is a content word or a function word. Without the need for specialized tools, this can be easily achieved via unsupervised clustering, namely predicting which K-Means cluster the query text belongs to.

## 3.3 Prompt

Prompt tuning is a promising paradigm that aims to adapt the knowledge from a pre-trained model to a target domain [15]. Mirroring the success of prompt tuning in natural language processing and cross-modal learning, we leverage it to facilitate CLIP for efficient text retrieval.

**Semantic-aware Prompting.** To improve retrieval performance on both content words and function words, we develop a semantic-aware prompting scheme. Inspired by CoOp [38], we introduce two sets of learnable context vectors to serve the retrieval of content words and function words respectively. Formally, the text prompts fed to the frozen CLIP language encoder are organized as:

$$P_c = [p_c^1, p_c^2, ..., p_c^M, Q] \tag{4}$$

$$P_f = [p_f^1, p_f^2, ..., p_f^N, Q] \tag{5}$$

where $P_c$ and $P_f$ denote the prompts for content words and function words respectively. $M$ and $N$ are the length of learnable context vectors, and $Q$ represents the query text.

Then, the CLIP language encoder outputs the prompt feature $f_{prompt}$. We calculate the cosine similarity between $f_{img}$ and $f_{prompt}$ to measure the pairwise similarity score between the input image and query text, i.e., $S(I, Q)$. The symmetric cross-entropy loss $\mathcal{L}_{align}$ over a batch is applied to contrastively align the matched $(I, Q)$ pairs.

**Distracted Queries Assistance.** Due to the limited input resolution and the visual-semantic entanglement, CLIP can perceive text to some extent, but it indeed lacks the ability of fine-grained character discrimination. To address this problem, a distracted queries assistance module is proposed, which teaches the FDP model to better recognize text during training. In particular, for a query text $Q$, we utilize a dictionary to generate $K$ (set to 5) distracted queries $Q^1, Q^2, ...Q^K$ that have the smallest edit distances with $Q$. These distracted queries are taken as hard negative samples which are also converted to text prompts and fed into the CLIP language encoder, predicting similarities scores $S(I, Q^k), k = 1, ..., K$. Meanwhile, the edit distance of each distracted query from the ground-truth query $D(Q, Q^k)$ is calculated. Subsequently, we convert these $K$ sets of similarity scores and edit distances into probability distributions and compute the KL divergence between them as the training loss $\mathcal{L}_{distract}$. The objective is to maximize the similarity scores of the distracted queries that are close to the ground-truth while minimizing the similarity scores of the distracted queries that are far from the ground-truth.

**Table 1: The setting of FDP models.**

| Model | Base model | Original input size | Expanded input size | $E$ |
|-------|-----------|--------------------|--------------------|------|
| FDP-S | CLIP-RN50 | 224 | 640 | 1024 |
| FDP-B | CLIP-RN50x4 | 288 | 720 | 640 |
| FDP-L | CLIP-RN50x16 | 384 | 960 | 768 |

## 3.4 Optimization

The proposed FDP is trained with the following loss functions:

$$\mathcal{L} = \lambda_1 \mathcal{L}_{loc} + \lambda_2 \mathcal{L}_{align} + \lambda_3 \mathcal{L}_{distract} \tag{6}$$

where $\lambda_1$, $\lambda_2$, and $\lambda_3$ are used to balance these losses, which are all set to 1 in our implementation.

During inference, the distracted queries assistance module is removed. Given a query text $Q$, images in the gallery are ranked according to the predicted similarity score $S(I, Q)$. When extending our FDP model to phrase-level or attribute-aware scene text retrieval settings, $Q$ is directly assigned the corresponding form of query, and the inference process remains unchanged.

## 4 EXPERIMENTS

### 4.1 Datasets

**IIIT Scene Text Retrieval (IIIT-STR)** [21] is a popular benchmark that contains 10000 images and 50 predefined queries. The images are collected using Google image search, so this dataset has a large variability in text fonts, styles, and viewpoints.

**Street View Text (SVT)** dataset [28] is a collection of natural street scenes. It consists of 100 training images and 249 testing images. All annotated words (427 words) in the test set are employed as the query text.

**TotalText** [4] is a scene text dataset consisting of 1255 training and 300 testing images. The 60 words that occur most frequently in the test set are selected as queries.

**Multi-lingual Scene Text (MLT)-Eng** is the English subset of MLT [22], which includes about 5000 images of natural scenes.

In our experiments, MLT-Eng is only used for training the proposed model. IIIT-STR, SVT, and TotalText are the testing datasets. It should be noted that as CLIP's potential is fully explored, 900k synthetic training images used in [27, 30] can be saved in FDP.

The query terms in existing datasets are all single words. To validate whether the STR models can be generalized to arbitrary-length query text, we introduce a new **Phrase-level Scene Text Retrieval (PSTR)** dataset. To build it, we select 36 phrases that occur frequently in life as queries, each containing 2 to 4 words, e.g., "*school bus*", "*handle with care*". For each query, we collect 15 images from TextOCR dataset [25] and Google image search respectively. In total, PSTR includes 1080 images and 36 query text.

### 4.2 Implementation Details

Based on CLIP with different capacities, we build several versions of FDP models, as summarized in Tab.1. As the input image size supported by CLIP is very limited, we expand the image size in FDP. However, directly expanding the image size makes the position embedding inherited from CLIP incompatible. To tackle it, we

**Table 2: Comparison with existing methods on IIIT-STR, SVT, and TotalText benchmarks. * means the result with subdivision enhancement. Bold indicates the best performance, and underline indicates the second-best performance.**

| Method | IIIT-STR | SVT | TotalText | FPS |
|---|---|---|---|---|
| Mishra *et al.* [21] | 42.70 | 56.24 | - | 0.10 |
| He *et al.* [9] | 46.34 | 57.61 | - | 2.35 |
| Jaderberg *et al.* [11] | 66.50 | 86.30 | - | 0.30 |
| ABCNet [16] | 67.25 | 82.43 | 69.30 | 17.50 |
| Gomez *et al.* [8] | 69.83 | 83.74 | - | 43.50 |
| Mafla *et al.* [19] | 71.67 | 85.74 | - | 42.20 |
| Mask TextSpotter v3 [13] | 74.48 | 84.54 | 72.42 | 2.40 |
| TDSL [27] | 77.09 | 89.38 | 74.75 | 12.00 |
| Wen *et al.* [30] | 77.40 | 90.95 | 80.09 | 11.00 |
| CLIP-RN50 | 52.93 | 65.07 | 38.46 | 76.32 |
| CLIP-RN50x4 | 52.60 | 70.54 | 41.65 | 57.91 |
| CLIP-RN50x16 | 53.03 | 76.55 | 43.51 | 29.02 |
| FDP-S (Ours) | 81.77 | 82.56 | 65.26 | **45.11** |
| FDP-B (Ours) | 86.65 | 86.64 | 73.63 | 31.43 |
| FDP-L (Ours) | 89.46 | 89.63 | 79.18 | 11.82 |
| FDP-L* (Ours) | **91.49** | **91.18** | **82.02** | 3.04 |

**Table 3: Ablation experiments on IIIT-STR and SVT datasets.**

| # | Focus | | Distinguish | Prompt | | IIIT-STR | SVT |
|---|---|---|---|---|---|---|---|
| | Dynamic Attention Shift | Text Knowledge Probing | | Semantic-aware Prompting | Distracted Queries Assistance | | |
| 1 | ✗ | ✗ | ✗ | ✗ | ✗ | 75.74 | 79.97 |
| 2 | ✓ | ✗ | ✗ | ✗ | ✗ | 78.38 | 80.21 |
| 3 | ✓ | ✓ | ✗ | ✗ | ✗ | 78.93 | 80.27 |
| 4 | ✓ | ✓ | ✗ | vanilla | ✗ | 80.07 | 81.03 |
| 5 | ✓ | ✓ | ✓ | ✓ | ✗ | 81.27 | 81.94 |
| 6 | ✓ | ✓ | ✓ | ✓ | ✓ | **81.77** | **82.56** |

propose a new learnable position embedding whose parameters are initialized with the nearest interpolation of original parameters.

We optimize FDP using Adam optimizer with an initial learning rate of 2e-3. The batch size is 48, and the number of training epochs is about 8. For fair comparisons, our experiments are implemented with Pytorch. All FDP models are trained on an NVIDIA A6000 GPU and tested on an NVIDIA 1080 GPU.

## 4.3 Comparison with Existing Methods

In this section, we compare FDP with existing methods on three STR benchmarks, *i.e.*, IIIT-STR, SVT, and TotalText. As a task to pursue practical applications, the inference speed of STR is undoubtedly very important, while previous methods are subject to the balance of retrieval accuracy and inference speed. In this paper, we first investigate employing the frozen CLIP model directly as the retrieval engine. As reported in Tab.2, it is surprising that CLIP already exhibits some retrieval capabilities even though it has not been specially trained on STR tasks. In particular, CLIP-RN50 obtains 52.93% and 65.07% mAP scores on the IIIT-STR and SVT datasets respectively, which even exceeds several dedicated STR models [9, 21] at a much faster speed (76.32 FPS).

Based on this observation, FDP is proposed to better unleash CLIP's potential for the STR task. On the IIIT-STR benchmark, we can notice that FDP-S initialized with the CLIP-RN50 base model boosts the mAP score by 28.84% (52.93%->81.77%), achieving an appealing result of 81.77%. Meanwhile, the inference speed is also superior (45.11 FPS), even faster than PHOC-based methods [8, 19]. When upgrading the model to a large size, FDP-L significantly outperforms the competitive model [30] by 12.06% (77.40%->89.46%) at a comparable speed. Compared with IIIT-STR, the query terms of SVT and TotalText contain many function words and often occupy small areas in images, which are more challenging for STR models. Nevertheless, even without complicated network design, FDP also reaches competitive performance on these datasets. To further boost the retrieval accuracy, we attempt to integrate the subdivision enhancement strategy here, *i.e.*, subdividing the input image into 4 equal patches and combing the outputs of these patches. The mAP scores are improved by 2.03%, 1.55% and 2.84% on IIIT-STR, SVT and TotalText, outperforming existing STR methods.

To provide intuitive analyses of FDP in comparison with previous methods, a typical example is visualized in Fig.6 (a). Given the query word "*adobe*", TDSL relies entirely on character composition

**Table 4: Analysis of the predefined probe on IIIT-STR benchmark.**

| Predefined probe | mAP |
|---|---|
| without predefined probe | 79.58 |
| "text" | 80.79 |
| "word" | 80.84 |
| "a set of text instances" | 80.41 |
| "scene text" | **81.77** |

**Table 5: Analysis of the context length on SVT benchmark.**

| $M$ ╲ $N$ | 2 | 4 | 8 |
|---|---|---|---|
| 2 | 81.80 | 81.71 | 80.65 |
| 4 | **82.56** | 82.11 | 81.68 |
| 8 | 82.01 | 81.44 | 80.86 |

for retrieval. If the scene text is blurry or small, it can easily be misrecognized. Besides, text-like patterns tend to interfere with model decisions. Instead, our FDP model takes full advantage of visual context information, returning the desired images from an image gallery. From the rank@7 and rank@10 images retrieved by FDP-S, we notice that the proposed method can recall images where the query text is not so salient.

### 4.4 Ablation Study

**Overall results.** In Tab.3, a detailed ablation experiment is conducted to verify the effectiveness of each module. We start by training a model that only utilizes the new learnable position embedding, whose mAP scores on IIIT-STR and SVT are 75.74% and 79.97% respectively. It reveals that enlarging the image size (*i.e.*, enhancing the text perceptual scale) is of critical importance for STR. Based on this, we gradually add the proposed modules and observe that each module brings noticeable improvements. In the "Focus" step, the dynamic attention shift and text knowledge probing modules can be considered to highlight scene text information from visual space and semantic space respectively. They bring 2.64% and 0.55% gains on the IIIT-STR dataset, which are proved to be effective. In particular, as IIIT-STR contains a large number of images without any text, the "Focus" step has a more significant effect on the IIIT-STR dataset than on the SVT dataset. Then, we study the effect of different prompt strategies. When simply adopting the learnable prompt method in [38] (#4), the mAP scores reach 80.07% and 81.03% on these two datasets. In contrast, we claim that content words and function words should be distinguished and exploit different customized prompts. Following this idea, our semantic-aware prompting scheme improves the performance to 81.27% and 81.94%. Further, by adding the training strategy of distracted queries assistance, 81.77% and 82.56% mAP scores are finally obtained.

**Analysis of the predefined probe.** The goal of the predefined probe is to stimulate the text-related knowledge hidden in CLIP. In Tab.4, we conduct an ablation study of the predefined probe on the IIIT-STR benchmark. The results show that if the predefined probe is removed, the mAP score decreases from 81.77% to 79.58%.

**Table 6: Analysis of the number of distracted queries on IIIT-STR benchmark.**

| $K$ | 3 | 5 | 7 | 10 |
|---|---|---|---|---|
| mAP | 81.51 | **81.77** | 81.75 | 81.61 |

**Table 7: Comparison with existing methods on PSTR dataset.**

| Method | mAP | FPS |
|---|---|---|
| Gomez *et al.* [8] | 68.01 | 42.45 |
| TDSL [27] | 89.40 | 11.34 |
| FDP-S (Ours) | **92.28** | **45.11** |

Furthermore, different strings are utilized to generate language embeddings that interact with the image attention feature. Compared to the "without predefined probe" baseline, these text-related probes can enhance the performance. Among them, "scene text" contributes to the best accuracy, implying that in CLIP's training data, the plain text "scene text" may appear frequently with the scene text content from images.

**Analysis of the context length.** In the semantic-aware prompting module, the hyperparameters $M$ and $N$ determine the context length for content words and function words respectively. In Tab.5, we evaluate the model performance on the SVT dataset to analyze the effect of these hyperparameters. According to the results, FDP reaches the best performance when $M = 4$ and $N = 2$. It may suggest that function words contain less semantics than content words, so they do not require complicated descriptions of context.

**Analysis of the number of distracted queries.** In the distracted queries assistance module, $K$ distracted queries are generated to help the model identify similar words. The ablation results of $K$ are reported in Tab.6. From them, we can see that a smaller $K$ may weaken the discrimination ability of FDP, while a larger $K$ will introduce many negative samples that are far from the query. Thus, $K$ is set to 5 in our experiments.

### 4.5 Extending to More Retrieval Settings

**Phrase-level scene text retrieval.** In reality, the scene text that people expect to retrieve is often of arbitrary length, such as "*ice cream*", "*do it yourself*". To validate the generality of our method over arbitrary-length query text, we evaluate FDP and several STR models on PSTR. For the comparison models [8, 27], since they can only accept word instances, we split each phrase into words, search them separately, and then average the corresponding similarity scores. As shown in Tab.7, FDP is more flexible than existing STR models in phrase-level retrieval. Specifically, the PHOC-based method [8] only achieves 68.01% mAP score on PSTR. We speculate this is because many split words are too short (*e.g.*, "*do*" in "*do it yourself*") to be accurately retrieved by the PHOC-based search algorithm. Although TDSL [27] can get 89.40% with the simple splitting operation, it is inherently flawed due to the local retrieval mechanism. From Fig.6 (b), we can see that for the query text "*no smoking*", TDSL may return images containing "*no engine*" (rank@3) or "*no softener*" (rank@5), which do not meet retrieval expectations. In addition, due to the dense text distribution in the PSTR dataset, these OCR-based comparison models run slower than on IIIT-STR. In

**Figure 6: Visualization of retrieval results. (a) An example on IIIT-STR benchmark, in which rank@6-10 retrieval results are provided. (b) An example on PSTR benchmark, in which rank@1-5 retrieval results are provided. The correct results are highlighted in green while the incorrect ones are highlighted in red. Best viewed in zoom.**

contrast, the FDP-S model reaches 92.28% mAP score on PSTR, outperforming existing methods by great margins. More importantly, as FDP does not rely on text detection or recognition processes, the retrieval speed will not be affected.

**Attribute-aware scene text retrieval.** Considering that people often query scene text with fine-grained attributes for more accurate search results, we further explore extending FDP to the attribute-aware scene text retrieval setting. We design some attribute-related queries and search corresponding images from the IIIT-STR dataset. Several typical retrieval examples are illustrated in Fig.7. These results manifest that the CLIP-based FDP model is naturally suitable for attribute-aware scene text retrieval because it takes advantage of CLIP's prior knowledge. FDP can well deal with attribute-related information such as color, font, and even position of scene text, returning images that users want. Admittedly, this is not available for conventional OCR-based STR models.

## 5 CONCLUSION

In this paper, we explore CLIP's intrinsic potential for efficient and flexible scene text retrieval. An OCR-free retrieval model named FDP (Focus, Distinguish, and Prompt) is proposed, in which the "Focus" design highlights scene text information hidden in CLIP while "Distinguish" and "Prompt" designs further overcome the negative effects caused by visual-semantic entanglement. Experimental results on three datasets demonstrate the effectiveness of our proposed modules and show that FDP achieves a better trade-off between retrieval accuracy and inference speed. In addition,

**Figure 7: Qualitative examples of attribute-aware scene text retrieval. Best viewed in zoom.**

FDP can easily generalize to other settings like phrase-level and attribute-aware scene text retrieval, which are more practical for requirements in real scenarios.

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
