# OpenReview forum: "Focus, Distinguish, and Prompt: Unleashing CLIP for Efficient and Flexible Scene Text Retrieval"
_acmmm.org/ACMMM/2024/Conference — MM2024 Poster_

### Official Review · Reviewer_dJfP · 2024-05-16

**Rating:** 3
**Confidence:** 4

**Summary:**

This paper aims to  retrieve the images using scene text based on CLIP model.

**Strengths:**

The strength is to explore CLIP for scene text image retrieval.

**Limitations:**

The work looks good but the following are the major weaknesses of the proposed work. (i) First of all exploring CLIP for scene text recognition and image retrieval is not new as the authors claimed in the paper. There are works that use CLIP for scene text prediction without recognition. The model uses image and textual features for prediction and text spotting in the scene images. There is no discussion on scene recognition and spotting in the paper as they are relevant to the proposed work. Once you recognize and spot the text in scene images, it is just adding a prompt step to retrieve. When we compare the proposed work with the state-of-the-art of scene text recognition and spotting, the novelty is limited. This is because the proposed work is just a combination of the existing models. The results are good but there should be comparative study with the latest existing methods for image retrieval. This is missing in the present form. At the same time, the authors must check the performance of the proposed method with latest CLIP based scene text recognition and spotting. In addition, there are methods for visual question answering. These methods work well for  any query. Again, there is no discussion about VQA methods. The authors must compare the proposed method with the existing VQA methods. Instead of using existing models for scene text detection and recognition, they should have developed their own method. Therefore, the contribution and novelty are limited.

Most important point is that the title says, it is a retrieval problem but the content show, it is a text detection and recognition problem. There are some results in the experimental section but there is no proper step presented in the proposed methodology section. After text recognition or spotting, how the method retrieves the images from the dataset is not clear. What is the difference between text detection, recognition, spotting and retrieval should  be clear in the work. There are many ways to retrieve the images. One way is to use detected text, another way is to use recognition results and another way is spotting. I did not see the proper method and the results to understand how the retrieval problem has been solved. When we split the whole phrase into words and then combine the score, how the method can assure the correct retrieval results. When we look at the retrieval results, there are images without text information. How an image without text can be retrieved. What is the advantage of text detection and recognition? Overall, it looks like the retrieval task is incomplete and there are not sufficient content in the proposed methodology section and the results in the experimental sections to justify the title.

**Suitability:**

2

---

### Official Review · Reviewer_QRr2 · 2024-05-23

**Rating:** 5
**Confidence:** 3

**Summary:**

The authors present FDP, short for Focus, Distinguish, and Prompt. FDP explores the potential of CLIP for scene-text retrieval, as previous work has proven that this model can recognize text (to some extent). However, CLIP is a vision and language foundation model and it is not explicitly trained to read. This means that, even though the model can perform scene-text retrieval, the entanglement between object and text features makes the retrieval results suboptimal. The three main parts of the pipeline (Focus, Distinguish, and Prompt) try to disentangle the object/background semantics and textual features and improve scene-text retrieval.

The "Focus" part of the network takes the image features from CLIP's vision encoder and highlights the text information of the image features, which results in a feature vector with richer textual features. The "Distinguish" part tries to help the model differentiate between content and function words, which further helps the disentanglement between visual and semantic concepts. The "Prompt" part processes the query (the text to be retrieved) and tries to align the ground truth image representation $I$ with the query representation $Q$. The authors also use a series of distractor queries $Q^k$ which they try to distance from the image representation $I$ and the query representation $Q$.

Furthermore, the authors introduce the PSTR dataset, a scene-text retrieval test set that features phrase-level queries, instead of words. The nature of the CLIP text encoder allows the authors to prompt the model with these phrases (or even whole sentences, but I imagine that is not as useful), as opposed to other retrieval models which can only process words.

**Strengths:**

The model appears to successfully adapt CLIP for scene-text retrieval. The different components of the network seem to have been carefully crafted to mitigate the biases that CLIP learned as a vision and language model. Even though plain CLIP already obtains decent scores (row 1 from Table 3), each component seems to help FDP focus on the textual features learned by CLIP.

The three versions of FDP (Small, Base, and Large) obtain better results in the IIIT-STR dataset than the SOTA, and the Small version manages to surpass all models in inference speed. However, the Small and Base models seem weaker in the SVT and TotalText datasets. The Large model that does not use subdivision seems to be competitive with [30], and the subdivision model obtains the SOTA in the three datasets, albeit at lower inference speeds.

I am especially interested in the results using phrases, instead of single words. Using CLIP's text encoder seems like a good way of expanding the scope and applications of scene-text retrieval. There is also uncovered potential in the "attribute-aware" retrieval, of which the authors only show qualitative results.

**Limitations:**

I am slightly worried about how much the image resolution helps the retrieval performance of FDP (a jump of 13\% on IIIT-STR and 14\% on SVT between CLIP-RN50 and base FDP-S from row 1, Table 3), a product of the limitations of CLIP when it comes to reading text. It makes me wonder if the failure cases of SVT and TotalText are mainly because of that.

I have some other minor concerns:

- The authors have used the ResNet-based image encoder from CLIP. Have the authors considered the ViT version?

- I would also appreciate a couple of qualitative examples showing the capacity of disentanglement of FDP compared to plain CLIP (model with the learned embedding vectors, row 1 from Table 3), although I understand that the space is limited.

- The Adam optimizer, which is used by FDP, is not cited in line 619.

- Finally, this is not as much of a concern, but I think the "attribute-aware"-based retrieval has the potential to be further explored in future works.

**Suitability:**

3

---

### Official Review · Reviewer_wZ7o · 2024-05-26

**Rating:** 4
**Confidence:** 4

**Summary:**

This paper explores the intrinsic potential of Contrastive Language-Image Pre-training (CLIP) for OCR-free scene text retrieval. Extensive experiments show that FDP significantly enhances the inference speed while achieving better or competitive retrieval accuracy.

**Strengths:**

The authors propose a novel model termed FDP (Focus, Distinguish, and Prompt) is developed. FDP first focuses on scene text via shifting the attention to the text area and probing the hidden text knowledge, and then divides the query text into content word and function word for processing, in which a semantic-aware prompting scheme and a distracted queries assistance module are utilized.
This paper is easy to understand. The description of the core idea is clear.

**Limitations:**

The notation in Table 2 is wrong.
The case study is not good enough.
It is better to pay more attention to show the superiority of OCR-free methods. The accuracy is not a good aspect.

**Suitability:**

3

---

### Official Review · Reviewer_nLc8 · 2024-05-27

**Rating:** 4
**Confidence:** 3

**Summary:**

The paper introduces a new CLIP-based method for scene text retrieval, aiming to find images containing specified query text without relying on OCR. The proposed pipeline uses pre-trained CLIP vision and text encoders, combined with a set of scene text retrieval scpecific trainable modules. First, a rough localization model on top of CLIP visual features and a multi-head cross-attention layer with the CLIP text ambedding for the phrase "scene text"  helps focusing attention on text areas. Second, they use K-Means over CLIP word embeddings to distinguish between different types of queries that leverage different learnable prompts as a kind of learnable query expansion mechanism. Finally, a distracted queries assistance module generates hard negative samples with minimal edit distances to the query and uses the KL divergence between CLIP similarity scores and edit distances as a training loss.

**Strengths:**

+ The proposed method achieves new state of the art results on several benchmarks.

+ The use of CLIP for scene text retrieval tasks is an unexplored approach that makes a lot of sense.

+ A new Phrase-level Scene Text Retrieval (PSTR) dataset is proposed, in which queries have more than on word.

**Limitations:**

- The paper does not clarify the source of the ground truth for training the rough localization model within the Dynamic Attention Shift module.

- The training data used for the KMeans module is not specified, and the decision to use only two clusters (k=2) is not well-justified. It is possible that using more clusters could yield better results, and this aspect should be further explored and discussed.

- The use of the Distracted Queries Assistance module implies modifications to the standard CLIP loss function, but the details provided are insufficient in my opinion. Specifically, it is unclear how the distractors for one query interact with the images relevant to other queries within a batch, and a more detailed explanation of the loss computation at the batch level is needed.

- The implications of distractors with occluded words in a given image, which might be relevant to the retrieval task, are not addressed. This oversight could impact the model's performance in practical scenarios where text occlusion is common.

- There are unclear statements in the Implementation Details section:
    "We expand the image size in FDP. We propose a new learnable position embedding whose parameters are initialized with the nearest interpolation of original parameters."
    "It should be noted that as CLIP’s potential is fully explored, 900k synthetic training images used in [27, 30] can be saved in FDP."

- The paper uses the RN50 version of CLIP rather than the ViT versions, which generally provide better results in other tasks. The rationale behind this choice is not explained, and comparing the performance of different CLIP versions could provide insights into the model's effectiveness.

**Suitability:**

3

---

### Meta-Review · Area_Chair_aLqc · 2024-06-30

**Recommendation:** Accept (Poster)
**Confidence:** 5

**Metareview:**

This paper aims to address the Scene Text Retrieval (STR) problem. The goal of STR is to return all images that contain the query text. The authors argue that most of the existing STR approaches conduct OCR to extract text from images and then conduct the similarity search, which could limit the efficiency. Towards this end, the authors propose an FDP model, which firstly leverages CLIP to roughly localize the text in the image, using k-means to split the query into ‘content’/ ‘function’, and explore the learnable semantic embeddings to enhance the query.

Most of the reviewers consider the application of CLIP in STR task interesting. The proposed FDP module can mitigate some of the drawbacks of CLIP in STR tasks, which enhances the technical contribution of this work.
There are a few issues that were pointed out by reviewers, including further clarifying experimental settings, discussion on failure cases of FDP, and typos. I would also request authors to report full results of “CLIP with large positional embedding”. As can be seen from table 3, it greatly improves the vanilla clip results and serves as a strong baseline. Importantly, the reviewer has commented that the paper should consider adding further related works on scene text recognition/spotting to strengthen the paper.

Balancing the strengths and weaknesses of this work, the overall sentiment among reviewers is positive. Given the innovative application of CLIP and the potential of the FDP model, I recommend a borderline acceptance, contingent upon the authors addressing the aforementioned areas for improvement.